# ICT Validation in Logistics Processes: Improvement of Distribution Processes in a Goods Sector Company

**Jose Alejandro Cano** [1,*]🆔**, Rodrigo Andrés Gómez** [2] **and Pablo Cortés** [3]

1 Faculty of Economic and Administrative Sciences, Universidad de Medellin, Medellin 050026, Colombia
2 Faculty of Administration, Politécnico Colombiano Jaime Isaza Cadavid, Medellin 050022, Colombia; ragomez@elpoli.edu.co
3 Escuela Técnica Superior de Ingeniería, Universidad de Sevilla, Camino de los Descubrimientos s/n, 41092 Sevilla, Spain; pca@us.es
* Correspondence: jacano@udem.edu.co

**Abstract:** This article aims to improve the secondary distribution process in a mass consumer company implementing technologies, such as transport management system (TMS) to achieve the objectives set by the company. A DMAIC based methodology is proposed to define and solve structured problems related to secondary distribution, following the performance of the process based on critical to logistics (CTL) factors. The methodology prioritized the design of a master plan for the secondary distribution and the characterization of the secondary distribution process, defining the principal technologies that should compose the business architecture of the secondary distribution, with emphasis on the TMS due to its significant impact and relevance for planning, execution, and control of the distribution process. This study replaces the control component of the DMAIC with the assess component to perform the economic and productivity evaluation of the implementation of a TMS since the improvement proposals were formulated and evaluated. The results show that TMS allows the reduction of delivery time variability, order processing time, voided invoices, distribution costs, the increase in customer service and efficiency in the distribution operation and generates profitability for the medium and long term.

**Keywords:** ICT validation; logistics; technologies; secondary distribution; DMAIC; TMS

## 1. Introduction

In today's business environment, companies need to develop a supply chain approach that allows them to synchronize and harmonize their logistics systems by considering flows of products, services, information, and finances between suppliers, producers, distributors, carriers, customers, and stakeholders [1–3]. Within the supply chain, the logistics distribution process allows the timely delivery of customer orders (fulfillment) considering strategies, means of transport, infrastructure, and information and communication technologies (ICT), thus meeting customer needs at the lowest cost [4,5]. Therefore, the planning, execution, and harmonized control of the distribution system, which includes the flows of information, services, products, and finances, are essential to satisfy the requirements of customers and stakeholders profitably at the lowest risk [6]. In this sense, the distribution process represents the set of logistics activities that mitigate the differences between demand and supply, confirming the availability of products among the actors in the supply chain [7]. Currently, this process is directly associated with the concept of city logistics, which considers conditions and restrictions for last-mile deliveries such as roads, mobility, and congestion, environment, quality of life of the inhabitants, among others, and in turn requires proper management of operations, facilities, transportation systems, inventory strategies, distribution networks, and ICTs [8,9].

Due to the rapid growth of cities and the number of users, the demand for urban freight transport continuously increases, expanding logistics flows in limited zones and

generating chaotic and uncoordinated transport [10]. Consequently, proper management of urban freight transport represents a fundamental element for sustainable economic growth [11]. It requires the distribution and transportation logistics systems to adopt ICTs to optimize the capacity of cargo vehicles, eliminate operational errors, and guarantee distribution routes that maximize performance, among others, therefore reducing costs, delivery times, delays, and pollution [12,13]. These conditions favor the use of logistics service providers (third-party logistics-3PL) as allies and partners in supply chains [14], pursuing a more efficient service to the end customer at a lower cost, adequate supply chain management, security in operations, and implementation and use of ICT [15,16].

The development of ICT is increasing, focusing more on digital platforms (portals, intranet/extranet), causing additional pressures for the competitiveness of supply chains [17]. The ICT involves directly with the customer relationships, manufacturing, transportation, warehousing networks, and data streams [18], mitigating problems related to goods payment, privacy and security, and traceability [19], and enhancing the firm's performance [20]. Consequently, ICT platforms represent one of the main pillars of smart logistics to improve intelligent transport systems [21] and support urban last-mile logistics through on-board telematics that allow real-time monitoring of vehicles and lay-by areas [22]. Likewise, ICT can reduce air pollution, congestion, and carbon emissions of the delivery sector through computerized routing, scheduling systems, and vehicle telematics systems [23,24]. In this regard, TMS impacts the reduction of $CO_2$ emissions within a distribution network by improving driving behavior, route optimization, and vehicle load level [25].

Likewise, the implementation of ICT in the distribution and customer service process is essential to support logistics management in real-time, allowing traceability and visibility of operations, resources, products, and actors, thus guaranteeing that deliveries comply with the quantity, quality, agreed delivery time, and established logistical conditions [26,27]. Therefore, ICTs are fundamental in the planning, execution, and control of distribution networks, since they allow the management of precise information in real-time, including the visibility and traceability of products, operations, and resources in the supply chain [27,28]. Investment in ICT has positively influenced productivity [29], allowing the supply chain execution (SCE) to optimize the movement of materials between distribution centers and other agents in the supply chain. Moreover, the success of the ICT implementation in logistics processes requires the proper selection and validation [30], the adoption of an appropriate ICT significantly impacts the competitive advantage of logistics companies and logistics operations [18], and companies require investments acquiring, updating, and maintaining technological infrastructure [31]. Therefore, it is necessary to review the appropriate methodologies to solve structured problems and support the selection and implementation of ICT for the secondary distribution.

Methodologies such as technology acceptance model (TAM), unified theory of acceptance and use of technology (UTAUT), dialogue–access–risk–assessment–transparency (DART), define–measure–analyze–improve–control (DMAIC), failure modes and effects analysis (FMEA), and quality function deployment (QDF) can predict the success of an ICT according to user perceptions, associated risks, and process modifications, ensuring ICTs become facilitators of value creation in logistics management. From these methodologies, the DMAIC is the most characteristic methodology of lean six sigma implementation in organization processes [32], is usually used to support and verify the ICT appropriation, solving structured problems, supporting management processes in ICT implementation projects [33,34], and guarantees a successful ICT implementation in logistics processes [30]. Likewise, the DMAIC methodology can be modified and supplemented, as in the case of DMADV, which employs the define, measure, analyze, design, and verify phases to replace existing systems with new processes [32].

Based on the research gaps in the literature, it is necessary to analyze the connections between lean six sigma (LSS) and ICT [35], extend the understanding of LSS implementation in specific contexts to create a background to support adopting LSS in logistics services environments successfully [36]. Likewise, it is required to produce more papers using

other already existing or completely new methods related to DMAIC [32] and conduct research on real organizations using ICT platforms [37]. Therefore, this study aims to improve the secondary distribution process in mass consumer companies through the use and appropriation of ICT, offering an improvement approach based on data analysis and management tools in supply chain logistics, especially for the distribution process improvement in terms of service level, freight costs on sales, and returns from customers. The main contributions of this paper are as follows:

- The DMAIC methodology improves the secondary distribution process in a company, involves the selection of ICT, and promotes connections between LSS and ICT in logistics.
- The DMAIC methodology is modified by discarding the control component since the improvement proposals are formulated, and it is replaced by the assess component to evaluate the economic and productive impacts generated in the secondary distribution system of a company.
- This study provides an application in a large food company to show the efficiency and effectiveness of the proposed methodology.

For this, Section 2 presents backgrounds on secondary distribution and methodologies for ICT selection and implementation, Section 3 describes the company under study, the methodology for the validation of ICT, and analyzes the information of the secondary distribution of the company based on critical to logistics (CTL) factors. Section 4 presents improvement opportunities in the distribution process considering a management approach and proposes a TMS to increase efficiency and customer service based on an economic and productivity analysis. Finally, Section 5 presents the conclusions and future works derived from this study.

## 2. Research Background

Companies dedicated to the manufacture and distribution of mass consumption food products typically serve distribution channels (stores and supermarkets) that must satisfy customer requirements (quantities, references, quality, and agreed delivery time) [38,39]. To achieve efficient distribution, these types of companies usually resort to primary distribution, which implies the distribution of products from the production facilities or central warehouses to the regional distribution centers (DCs), and a secondary distribution involving deliveries from regional DCs to customers (supermarkets and stores), relating directly to city logistics models [40,41].

The secondary distribution presents more considerable organizational and logistical complexity than the primary distribution due to stores and supermarkets located in high-density urban areas. It implies managing congestion, restrictions for certain types of vehicles in delivery areas, driving regulations, reduced speed, while the distances between delivery points are usually relatively short [9,13,42]. Additionally, the structure of the urban distribution network typically contains multiple origin nodes (factories or warehouses), destination nodes (warehouses, stores, or supermarkets), and arcs connecting nodes (distances, costs, or times between nodes). Therefore, it is necessary to implement quantitative models in distribution systems to design routes, comply with the planned service levels, and achieve the highest efficiency in the distribution network [8,43].

These solutions are usually framed in ICT; however, the rapid development of technologies increases the number and scope of the tools applicable to several logistics processes, causing an increase in the ICT available for logistics processes, which impedes decision making about which technology is more convenient to implement [31]. From these technologies, TMS has fulfilled a strategic role in the planning and management of supply chain activities due to the importance of transportation management in the costs and interactions in companies, customers, and suppliers [44], including integrative digital technologies in the supply chain aligned with logistics 4.0 [45] and geographic information systems that support decision making, traceability, and route monitoring for distribution and transportation of goods [46].

TMS is the ICT that continuously optimizes transport variables by increasing efficiency, improving customer service, and reducing distribution costs. TMS connects supply chain agents and generates visibility to shipping and order fulfillment activities automating transportation operations. Similarly, a TMS supports the transportation processes from the operational execution and the financial settlement, allows operations traceability, freight payments, managing indicators, making reports, and scheduling docks, offering a range of solutions to the planning and execution of transport, freight management, foreign trade, package delivery, and connectivity. The scope of the TMS in operations planning determines the minimum routing cost to serve a customer and determines from which warehouse the orders should be delivered to customers [44].

TMS has been approached from the scientific literature by different authors who highlight the importance of effective management of supply chain technologies, especially between retailers and suppliers, to obtain satisfactory joint performance in relationships, collaboration, and synchronization between the parties [47]. Asimakopoulos et al. [48] developed the DynaCargo (dynamic cargo routing on-the-go) project for sustainable ecological development in urban spaces implementing a TMS for urban solid waste collection, relying on RFID units, DTN network protocols, and dynamic routing algorithms to minimize hardware and telecommunication costs. Similarly, Evensen [34] works with an ATMS (advanced transportation management systems), which combines transportation management services, such as intersection control, pollution reduction, access control, management of parking lots, docks, ramps, among others, and enables customized services. On the other hand, Mu [49], Li et al. [50], and Ashour et al. [51] propose TMS applications for the transport of passengers, through transport routes, vehicle monitoring, bus stations, implementation of GPS (global positioning system) or GPRS (general packet radio service) applications and integration of systems of urban and rural transport. Karoń and Janecki [52] analyzed the development of an intelligent transport system (ITS) for a city in Poland.

However, it is necessary to use a technology validation methodology to determine the relevance of an ICT for a logistics process. Based on [30], Table 1 presents a comparison between methodologies used for the validation of ICT for logistics processes, highlighting the advantages and disadvantages of each one. From these options, the DMAIC is chosen, which despite representing a generalist and not a very standardized methodology, it can be adapted and modified according to the requirements of the logistics process. Likewise, DMAIC is used because it is a comprehensive methodology that seeks the continuous improvement of processes, focusing not exclusively on establishing whether the ICT is relevant or not but also on determining the requirements to overcome identified problems. Accordingly, in this study, the control component is replaced by the assess component because the proposed scope covers up to the economic and productive validation of the solutions.

To guarantee a structured improvement within logistics processes, methodology, such as DMAIC, is required to provide a framework to structure robust and high-quality solutions according to problems faced by companies [53]. Consequently, LSS represents a data-driven approach, which uses specific methodologies to reduce variability, eliminate waste within business processes, and lead to fact-based decisions through statistical tools and techniques. DMAIC consists of understanding the root causes of a problem through exhaustive analyses to propose assertive solutions and prioritize premature decision making to improve the process [54], representing one of the methodologies ensuring the implementation of supply chain management practices in companies [55,56], reducing security risks in container transport within the supply chain [57], developing improvement models for the analysis and interpretation of performance indicators [58], improving logistics replenishment processes, the efficiency of the transportation process, and business performance in terms of on-time in-full orders, quality level in the supply chain network [59].

**Table 1.** Advantages and disadvantages of ICT validation methodologies.

| Methodology | Advantages | Disadvantages |
|---|---|---|
| DART | It allows technology providers to involve customers as collaborators, facilitates dialogue with consumers and risk assessment on both sides. It provides the developer with consumer expectations and experiences to improve trust. | It was created for generic products or services and requires intense interaction with each consumer. Discussing options does not necessarily give customers a degree of control over the responsibilities to assume. |
| DMAIC | It is used as a continuous improvement method for understanding the root causes of a problem, provides procedures for effective integration of tools within a systematic framework, and includes powerful statistical techniques for hypothesis verification. | The generality of the method. The identification of the causes of potential problems has no strategic orientation. It does not use simulation and optimization tools to model complexity. |
| QDF | It can transform the customer's needs into technical solutions to improve the performance of a process, covering all the development stages of a technology. | Information about individual judgments can be generated in multiple formats that may be alien to the knowledge of the individual. The preferences generated can be difficult to assess consistently. |
| TAM | Model for predicting the use of information and communications technologies. Effective alternative to analyze the reasons that lead individuals to adopt new technologies. Simplicity, adaptability, and theoretical strength. | Model dependent on external factors that can be diverse like cultural factors. Lack of relationship among psychological, social, and contextual variables such as material access conditions or digital user skills. |
| UTAUT | It helps to understand the acceptance factors during the proactive design of technologies, aimed at users less likely to adopt and use new systems. | The limited application for some business areas. Most of the studies performed have been within the same country, discarding culture as a technology acceptance criterion. |

DMAIC methodology provides labor costs savings, eliminates replenishment routes without generating negative impacts in the process, and increases employee and customer satisfaction. It entails improving transport and distribution logistics processes to lead them to world-class processes. Therefore, it is pertinent to use the DMAIC methodology in this study since it has proven to be a successful methodology to improve logistics processes, especially in transportation and distribution systems, and can complement other methods for improving business processes [56]. DMAIC can be used in various sectors because it is not a standardized procedure and is used as a continuous improvement method [32]. It has been used as a framework for supporting continuous improvement in logistics processes, encompassing logistics services [36], procurement processes with ICT for process digitization purposes [35], reverse logistics processes for the pharmaceutical industry [60], inbound logistics and production scheduling to manage the transformation from traditional operations towards the integration of the IoT and cyber-physical systems [61], service and delivery processes to increase the level of compliance with service agreements, reduce CO2 emissions, improve invoice processing time and delivery fill rates [62], supply chain strategies in collaboration with ICT platforms to make organizations resilient to disruptions [37], internal logistics and hospital sustainable digital transformation to elaborate an intelligent support system [54], and supply chain management to minimize lead times focusing on the improvement in processes, information systems, organizational structures, and advances in distribution and transportation technologies [63].

## 3. Materials and Methods

This study proposes a methodology based on some DMAIC components (definition, measurement, analysis, and improvement) and tools for the logistics process improvement in the supply chain to achieve the objectives of a secondary distribution system. The control component of the DMAIC is discarded since the improvement proposals will be formulated and evaluated. Additionally, this study includes a methodology component called economic and productivity evaluation to assess the potential impacts generated

by the proposed improvements for the secondary distribution system in the company. DMAIC methodology represents a systematic analytic procedure that could be generalized to a wide range of applications [64]; therefore, the methodological approach based on the DMAIC is appropriate since it allows for developing improvement proposals based on the data analysis from the secondary distribution network using statistical techniques and process management. Figure 1 shows the stages of the methodological framework to achieve the objective of this study.

| DEFINE | MEASURE | ANALYZE | IMPROVE | ASSESS* |
|---|---|---|---|---|
| **Stage 1:** Definition of the secondary distribution problem | **Stage 2:** Secondary distribution process measurement | **Stage 3:** Analysis of the secondary distribution process | **Stage 4:** Improvement of the secondary distribution | **Stage 5:** Economic and productivity evaluation* |
| Definition of the logistics problem of secondary distribution in the supply chain of the company under study. Scope of the improvement project. Project work planning for the secondary distribution improvement. | CTL identification and description. The information is obtained mainly from ERP reports. Primary data is obtained from surveys to establish the structure of the process and the perception of the process staff. | Data analysis using descriptive statistics to establish the performance of the logistics process. Identification, and prioritization of logistics factors causing performance. Improvement opportunities. | Formulation of improvement opportunities to increase the performance of the secondary distribution considering objectives, CTL variables, scope, and relationships with other supply chain processes. The technical structure of the proposals. | Productivity improvement analysis considering the increase in customer service and average returns. Economic evaluation including costs reduction, return on investment (ROI), payback period, and profitability. |

*\* The CONTROL component of the DMAIC is discarded since the improvement proposals will be formulated and evaluated. It is replaced by the ASSESS component, used for the economic and productivity evaluation to assess the potential impacts generated by the proposed improvements.*

**Figure 1.** Stages of the methodological proposal.

The company under study belongs to the food sector (refrigerated and frozen food), producing more than 1200 tons of products per month and developing high-quality processes. The company's portfolio exceeds 400 references between its brand and brands manufactured for third parties. The company promotes products through various distribution channels: chain channels, supermarkets, mini markets, direct store delivery (DSD), institutional sales, and state contracts. Through nine distribution centers (DC), the company can distribute its products directly to the customer. However, in recent years, the company declared losses due to congestion caused by the inadequate physical infrastructure in the operating cities, the competitive issues in the market, and problems related to the installation and appropriation of comprehensive management software that caused significant trauma to the company. In addition, some logistics problems related to quality, quantity, and delivery timeliness persist, for which it is necessary to improve the service level, the freight cost, and the order returns indicator.

This section introduces the first three stages of the methodological framework, including the definition of the distribution problem (Stage 1), measurement of the distribution process (Stage 2), and analysis of the distribution process (Stage 3).

### 3.1. Definition of the Secondary Distribution Problem (Stage 1)

This stage delimits the company's problem. Table 2 establishes the scope of the problem and shows a deficiency in the service level by being 9.5% below the set goal. Similarly, cost inefficiencies due to returns and voided invoices severely affect EBITDA (earnings before interest, taxes, depreciation, and amortization) and profitability. Therefore, the secondary distribution is affecting the service level and the efficiency of the company and its supply chain.

**Table 2.** Definition of the secondary distribution problem.

| Problem Statement | | |
|---|---|---|
| The company has designed a secondary distribution model allowing to serve more than 10,000 stores and supermarkets nationwide. A logistics analysis established that the company presents low logistics efficiency in the secondary distribution of its nine DCs. Five logistics variables reflect this fact. | | |
| **Indicator** | **Current Value** | **Desired Value** |
| Secondary freight | 6.2% of annual sales | 5.0% of annual sales |
| Voided invoices | 0.45% of annual sales | 0.3% of annual sales |
| Product quality returns | 1.29% of annual sales | 1.0% of annual sales |
| Product turnover returns | 2.34% of annual sales | 1.0% of annual sales |
| Service level (Depends on the above indicators) | 85.5% | 95% |
| **Project Scope** | | |
| The project covers the secondary distribution process and the interaction with other logistics processes. The project begins with secondary data analysis for the logistics variables, performing a diagnosis of the secondary distribution. Based on the information analysis, improvement projects are proposed to solve the deficiencies and increase the service level and operational efficiency. | | |

### 3.2. Measurement of the Secondary Distribution (Stage 2)

This stage measures the secondary distribution, beginning with the definition and description of the CTL factors. CTL factors represent variables that directly impact the secondary distribution process and must be under control to visualize the effectiveness of the process. For this case, the CTL factors will be the variables on which the quantitative and qualitative study will be focused to understand the current problem. Table 3 presents these quantitative variables.

**Table 3.** Study variables.

| Variables | Importance of the Variable |
|---|---|
| Secondary freight | Indicates the annual expenses in the secondary distribution. |
| Product quality returns | It involves additional transportation costs. It reflects the efficiency of logistics planning. |
| Product turnover returns | It involves additional transportation costs. It reflects the efficiency of logistics planning. |
| Voided invoices | Indicates lack of communication and negotiation, representing expenses in transport and storage. |
| Service level | Indicates the units billed and delivered correctly. They represent the success of the operation. |

The information collected for each CTL factor is obtained from historical data reports from the company's ERP. The primary information comes from a survey directed to the logistics staff of the company. This information is arranged according to dates, regional DCs, and CTL to establish the current performance measurement of the secondary distribution. Different descriptive statistics techniques provide a quantitative approach to guarantee the validity of the analysis of the distribution process.

### 3.3. Analysis of the Secondary Distribution (Stage 3)

This stage develops a descriptive diagnosis and a quantitative performance measurement for the secondary distribution to understand the characteristics and performance of the process. A survey applied to the logistics management and the collaborator in this logistics provides primary data. The questions are related to the CTL factors defined for the secondary distribution and are listed below:

- How many DCs does the company have in the supply chain?
- How is the secondary distribution process performed in the company?
- Are the cargo vehicles owned by the company or subcontracted?
- What are the main criteria for selecting a logistics operator?

- How are freights managed in the distribution of the company?
- Which of the two freights (primary or secondary) is more important for the company and why?
- Are there cost overruns in the company's secondary transport?
- Is an ICT implemented in the company for distribution and transportation management?

The information obtained from the survey indicates that secondary distribution directly impacts customer satisfaction and profitability. The secondary distribution is highly complex since several municipalities are served from nine regional DCs and a DC attached to the production plant. The secondary distribution process is developed in several phases in which the company, the customer, and the carrier interact. These phases include the storage of the finished product in regional DCs, and for this, the primary freight is transported in large capacity trucks (large volumes) from the DC of the production plant to regional DCs. The sellers then record the orders in the Enterprise Resource Planning (ERP) system so that the finished product can be invoiced based on the seller's orders. Each DC prepares the dispatch order, and the carriers distribute the products from the regional DC to the customers. Since the company does not own a vehicle fleet to transport the products from the DC to the points of consumption, the transport of the secondary freight is subcontracted with cargo transport companies (3PL) that fulfill the specifications required by the company (logistical capacity, financial capacity, national coverage and offices in the DC locations, competitive costs, and Money Laundering Risk Management System approval).

On the other hand, the secondary freight cost is relevant (approximately 7% on sales) due to inefficiencies generated by lack of synchronization and collaboration with other actors in the supply chain, especially with transporters and customers, as well as low-effectiveness logistics and commercial practices. Likewise, the company has an ERP system that provides an administrative and commercial approach to orders, but not a logistics approach that supports management, traceability, and visibility in the secondary distribution (routing, traceability, visibility, and transportation and distribution management), and hence inefficiencies are generated in productivity and costs described above.

Regarding the quantitative measurement of performance, an analysis is performed for the five CTL factors of the secondary distribution: secondary freight, voided invoices, product turnover returns, product quality returns, and service level.

### 3.3.1. Secondary Freight

Table 4 shows the secondary freight costs by regional DC and their share of the total costs of secondary freight, where DC 1, 2, and 3 represent 58.9% of the total expenses in secondary transportation per year. Logistics improvement strategies must focus on these DCs to improve the secondary distribution since they compile the highest sales and transportation operations.

**Table 4.** Secondary freight share.

| DC | Secondary Freight (USD) | % Share | % Accumulated |
|---|---|---|---|
| DC 1 | 466,981 | 29% | 29.30% |
| DC 2 | 268,842 | 17% | 46.20% |
| DC 3 | 202,402 | 13% | 58.90% |
| DC 4 | 164,415 | 10% | 69.30% |
| DC 5 | 153,725 | 10% | 78.90% |
| DC 6 | 109,880 | 7% | 85.80% |
| DC 7 | 90,255 | 6% | 91.50% |
| DC 8 | 63,372 | 4% | 95.50% |
| DC 9 | 53,916 | 3% | 98.80% |
| DC Production Facility | 18,447 | 1% | 100.00% |
| Total | 1,592,236 | | |

Table 5 shows the consolidated annual freight costs on sales, indicating a sales share of 6.15% for the secondary freight, representing a significant value concerning the goal set by the logistics management (5% on sales). Therefore, the company requires improvement strategies for its reduction since this CTL factor impacts the profitability and efficiency of the secondary distribution.

**Table 5.** Consolidated annual freight.

| Total Annual Sales | Annual Primary Freight (USD) | % On Sales | Annual Secondary Freight (USD) | % On Sales |
|---|---|---|---|---|
| $25,899,320 | 477,556 | 1.84% | 1,592,236 | 6.15% |

The capacity of the trucks and their respective percentage of use must be analyzed since the trucks used in the secondary distribution must dedicate 25% of the volume to allow air circulation in the cold chain and guarantee logistics operational excellence. Table 6 shows the average truck utilization for each regional DC, identifying that none of the regional DC uses 75% of the truck capacity to perform the secondary distribution. The DC with the highest truck utilization is DC 1 with 68%, while the lowest is DC 9 with 33% on average. According to the weighted average of vehicle use, considering the participation of each DC in sales, the company presents an average vehicle occupancy of 61%. These results highlight the need to implement logistics improvements to increase truck utilization and increase logistics efficiency.

**Table 6.** Percentage of truck utilization by regional DC.

| DC | % Capacity Utilization |
|---|---|
| DC 1 | 68% |
| DC 2 | 58% |
| DC 3 | 57% |
| DC 4 | 69% |
| DC 5 | 59% |
| DC 6 | 63% |
| DC 7 | 49% |
| DC 8 | 66% |
| DC 9 | 33% |

### 3.3.2. Voided Invoices

Voided invoices represent orders not delivered to the customer even though they have been invoiced. The cancellation of invoices impacts the efficiency and profitability of logistics, hence its importance in the present analysis. The information collected reflects that 80% of the voided invoices during a year belong to DC 1, 3, 4, 5, representing a value of USD 592,987, equivalent to 2.3% of the company's annual sales. Tools such as cause–effect diagrams indicate that the most critical causes relate to logistics, internal, and external communication problems and can be improved by implementing real-time ICTs and training the staff to renegotiate with customers. Regarding transport logistics problems, the analysis identifies issues, such as customers unattended by drivers, non-compliance in delivery times, and driver's delivery errors, representing an annual value of USD 115,399. These problems can be mitigated using efficient and specialized ICTs for transport management.

### 3.3.3. Product Turnover Returns

Product turnover returns represent customers maintaining the product at their facilities, and it is necessary to pick up as it approaches expiry because the sale has yet to be attributed to the final consumer. It occurs due to an error in the demand forecast, and the company takes care of the reverse logistics for its collection. The procedure of returns

is relatively similar during all months, and they are related to low sales, poor inventory management, short expiration dates, among others. The logistics cost of returns represents an average annual value of USD 53,959. These values indicate the importance of improving this CTL factor since returns do not generate value for customers and affect the efficiency and profitability of the supply chain.

### 3.3.4. Product Quality Returns

Product quality returns occur when some characteristics of the delivered product are not complying with the quality specifications, generating risks to the consumer's health. Therefore, the product is collected and later destroyed or sold as raw material for other industries. These returns affect operational efficiency, customer satisfaction, and profitability. The cost of these returns represents an average of 1.29% of annual sales, while the company's goal is to obtain 1% of product quality returns.

### 3.3.5. Service Level

The service level is measured by comparing the units billed with the units requested by customers. The service level decreases with each line that has been ordered and has not been invoiced. The secondary distribution of the company impacts the service level since it is responsible for delivering the products to the customers. From Table 7, it is deduced that the service level is on average at 85.5%, which is lower than the established goal of 95%, and that no regional DC reaches it, highlighting the need to implement an improvement plan to increase it.

**Table 7.** Service level by regional DC.

| DC | Service Level |
| --- | --- |
| DC 1 | 90.7% |
| DC 2 | 81.5% |
| DC 3 | 78.7% |
| DC 4 | 86.0% |
| DC 5 | 84.8% |
| DC 6 | 88.3% |
| DC 7 | 85.9% |
| DC 8 | 91.7% |
| DC 9 | 81.7% |

Therefore, the company must improve the service level, reduce the costs of secondary freight, reduce the cancellation of invoices and customer returns. Likewise, the company does not meet the goals set for the five CTL factors, which leads to the formulation of improvement proposals for the secondary distribution in the company. Moreover, the logistics strategies and processes associated with the secondary distribution are not strategically aligned with the conditions and restrictions of the city logistics where the company's DCs are located. Consequently, the company requires improvement proposals for the secondary distribution considering the collaboration and synchronization of logistics processes within a city logistics approach. Likewise, it demands the implementation of ICTs with real-time functionalities. The improvement of the secondary distribution of the company will allow the appropriation of ICTs that have been developed in the scientific literature and world-class operations, facilitating the approach to the concepts of city logistics.

## 4. Results and Discussion

This section introduces the last two stages of the methodological framework, including secondary distribution improvement from a management approach (Stage 4) and the economic and productivity evaluation for secondary distribution improvements (Stage 5).

### 4.1. Secondary Distribution Improvement from a Management Approach (Stage 4)

Based on the performance analysis of the CTL factors, it is necessary to identify and select improvement opportunities to meet the goals set for the secondary distribution. The improvement opportunities are detailed in Table 8, describing the factor, current problem, proposed solution, and expected impacts.

**Table 8.** Improvement opportunities for CTL factors.

| Factor | Problem | Solution Proposal | Impact |
|---|---|---|---|
| Voided invoices | They represent 2.3% of annual sales. Products associated with voided invoices cannot always be resold and sometimes are transferred to other DCs for marketing. | Confirmation strategy with customers through the sales force. Incentive program for sellers with the least number of voided invoices per month and incentives for customers. | Reduction of the indicator by at least 1% on the company's global sales. Reduction of logistics costs associated with the products of the invoices. |
| Secondary freight | Secondary freights represent 6.15% of annual sales. These freights include the transport of products associated with Voided invoices. The vehicles maintain an average occupation of 61%, being 14% below the maximum capacity. | Synchronize sales with logistics to improve the utilization of vehicles. Implement a cubic capacity model to increase the used capacity of the trucks. Design a mathematical model for vehicle routing to reduce travel times and increase the coverage of customer areas. | Increase the average used capacity of trucks by 7%, achieving an average utilization of 68%. Reduce the cost of secondary freight over sales to 5.5%. |
| Product turnover returns | They evidence inadequate management in the sales force, poor inventory management, and poor management of the FIFO system. It implies primary and secondary transportation costs in reverse logistics. | Provide the seller real-time information to identify sales history and suggest the quantities to be delivered, reducing the amount of returned product. Implement demand forecast analysis to provide strategic decisions about customer shipments. | Reduce the indicator by 1%, reducing the impact on reverse logistics costs and secondary transport. |
| Product quality returns | They represent 1.29% of annual sales and occur when the product presents quality problems or loss the cold chain due to mishandling by some customers and DCs. The product is brought back to the production facility, while other products are sent back to fulfill the sales promise. | Train customers regarding refrigeration management. Provide refrigeration systems to customers with large purchase volumes. | Reduce product quality returns to 0.5% on sales. |
| Driver does not visit customer | 15% of voided invoices are because drivers do not visit the customer due to difficult access to stores or mini-markets, social problems, armed conflicts, or simply drivers evade routes and do not deliver the product. This phenomenon generates duplication of trips, waste of capacity in the truck, and non-fulfillment of orders. | Redefine the freight table by incentivizing drivers based on the number of perfect orders. Establish a minimum percentage of perfect orders per driver, which must be evaluated periodically. Apply non-compliance sanctions to drivers. Implement route traceability and customer service system. | Reduce the indicator by at least 1%. |

**Table 8.** *Cont.*

| Factor | Problem | Solution Proposal | Impact |
|---|---|---|---|
| Secondary distribution management | The strategies of the secondary distribution process are unsynchronized with the logistics conditions of the cities. It increases the secondary freight costs, prevents customer service compliance, and generates returns. | Develop a master plan to synchronize the supply chain considering objectives, goals, indicators, ICTs, and other components. Redesign the secondary distribution process based on the results of the master plan. Appropriation of a TMS aligned with the company's WMS and ERP to support the secondary distribution. | Increase the service level to at least 95%. Reduce total returns to a maximum of 2% on sales. Reduce freight on sales at a maximum of 5%. |

A structured dialogue methodology was implemented to select the improvement opportunities, in which the logistics manager, a group of internal analysts, and external experts analyzed the improvement opportunities. This group assigns a score from 1 to 10, where 10 represents the highest rating. Table 9 presents the scoring results, identifying that the secondary distribution management, secondary freight, and product turnover returns received the highest rating (greater than 8.5).

**Table 9.** Rating of improvement opportunities for secondary distribution.

| Improvement Opportunities | Rating |
|---|---|
| Secondary distribution management | 9.5 |
| Secondary freight | 9.0 |
| Product turnover returns | 8.9 |
| Voided invoices | 7.3 |
| Product quality returns | 7.1 |
| Driver does not visit customer | 7.0 |

Therefore, the improvement proposals consist of improving the secondary distribution from a management approach and implementing a TMS (transport management system) to support the effective management of secondary distribution. The management approach is based on a master plan that establishes goals, strategies, logistics processes, organizational structure, and ICTs such as ERP, order management system (OMS), warehouse management system (WMS), and TMS. This master plan (see Table 10) must be aligned with the strategic plan of the company and its supply chain strategies.

**Table 10.** Master plan for the secondary distribution.

| Objective | | | | | |
|---|---|---|---|---|---|
| Manage the secondary distribution of the company to achieve a minimum service level of 95%, percentage of freight costs over sales at 5%, and maximum nationwide returns of 2%. | | | | | |
| **Scope** | | | **Responsible** | | |
| Covers from the dispatch operation in regional DCs to the delivery orders to customers and reverse logistics of returns | | | Logistics Manager | | |
| **Goals** | **Process** | **Indicator** | **Team** | **Resources** | **ICT** |
| Guarantee a customer service level of at least 95% per month. | Secondary distribution | % Service level per month | Logistics manager, Distribution analysts, DC collaborators | DC infrastructure 3PL services | ERP, OMS, WMS, TMS |

**Table 10.** *Cont.*

| Objective | | | | | |
|---|---|---|---|---|---|
| Obtain customer returns for logistical conditions and maximum product quality of 2% per month. | Secondary distribution | Monthly returns due to logistics conditions | Logistics manager, Distribution analysts, DC collaborators, 3PL | DC infrastructure 3PL services | ERP, OMS, WMS, TMS |
| Ensure that freight on sales is a maximum of 5% in regional DCs. | Secondary distribution | % Freight on sales per month | Logistics manager, Distribution analysts, DC collaborators, 3PL | 3PL services | ERP, TMS |
| Generate at least 95% perfect deliveries to customers. | DC regional (order picking and shipment) Secondary distribution | % Perfect deliveries per month | Logistics manager, Distribution analysts, DC collaborators, 3PL | DC infrastructure 3PL services | ERP, OMS, WMS, TMS |
| Guarantee the occupation of trucks in the secondary distribution is on average between 70 and 75%. | Secondary distribution | Average truck occupancy per month | Logistics manager, Distribution analysts, DC collaborators, 3PL | 3PL services | ERP, TMS |
| Provide more than 95% of effective deliveries per month. | Secondary distribution Sales | % of effective deliveries per month | Logistics manager, Distribution analysts, DC collaborators, 3PL, Regional sales manager | 3PL services | ERP, OMS, WMS, TMS |
| Generate a maximum of 5% of voided invoices per month. | Secondary distribution Sales | % of voided invoices per month | Logistics manager, Distribution analysts, DC collaborators, 3PL, Regional sales manager | Invoice information associated with deliveries | ERP, OMS, WMS, TMS |

Table 11 presents the characterization of the secondary distribution based on a SIPOC analysis (suppliers, inputs, process, outputs, customers), including improved activities, and the proposal to appropriate real-time ICTs to support the decision-making process. On the other hand, the characterization is aligned with the master plan and the logistics strategy. The most relevant improved activities are the planning and selection of means of transport according to the logistics conditions of the delivery, as well as routing planning considering minimal tardiness in the secondary distribution network. Additionally, the characterization includes the TMS implementation to support the management of the secondary distribution, including the synchronization with the WMS and the ERP.

**Table 11.** Characterization of the secondary distribution.

| Objective | | | | |
|---|---|---|---|---|
| Manage the secondary distribution of the company to achieve a minimum service level of 95%, percentage of freight costs over sales at 5%, and maximum nationwide returns of 2%. | | | | |
| **Scope** | | | **Responsible** | |
| Covers from the dispatch operation in regional DCs to the delivery orders to customers and reverse logistics of returns | | | Manager of each regional DC | |
| **Suppliers** | **Inputs** | **Process (Activities)** | **Outputs** | **Customers** |
| Sales force | Customer order information (reference, quantity, delivery date and logistics conditions) | Take customer orders using the order management system | Registered customer order | Shipment planning |

**Table 11.** *Cont.*

| | | **Objective** | | |
|---|---|---|---|---|
| Shipment planning | Consolidated customer orders for each regional DC | Extract the order master for each regional DC that serves customers in the area | Order master classified by days and customers for deliveries | Shipment planning |
| Shipment planning | Customer order master classified by delivery logistics conditions | Upload the order master to the ERP for synchronization with other logistics and administrative processes of the company | Consolidated orders loaded in the ERP | Shipment planning |
| Shipment planning | Consolidated orders loaded in the ERP | Validate restrictions on information uploaded to the ERP. Date of dispatch, order information, time restrictions, customer information, city logistics conditions. | Refined orders to be delivered to customers | Shipment planning |
| DC | Consolidated picking orders by route in regional DCs | Check inventory availability and manage order exceptions in regional DCs using WMS | Confirmation and assignment of stocks in the WMS aligned with the ERP | Shipment planning |
| Shipment planning | Refined orders to be delivered to customers | Design the routes of secondary distribution using a TMS to minimize travel time | Consolidated picking orders by route in regional DCs | Shipment planning |
| Shipment planning | Consolidated picking orders by route in regional DCs | Assign vehicles to the routes using the TMS | Picking orders consolidated by route and vehicle | DC |
| Shipment planning | Confirmation and assignment of stocks in the WMS aligned with the ERP | Generate picking orders by route using the WMS | Picking orders | Shipment planning |
| DC | Picking orders | Load configuration to guarantee the adequate occupation of the trucks (70 to 75%) | Load configuration | Billing |
| Billing | Load configuration | Generate invoices by distribution route | Load sheet and invoices by route | DC |
| DC | Vehicle loading | Certify the vehicle load to guarantee quantities, references, and quality | Load form with acknowledgment of receipt | Route validator Carrier |
| DC | Delivery scheduling | Deliver orders by executing the routes designed in each zone | Signed bills | 3PL |
| 3PL | Signed bills | Consolidate deliveries per day by synchronizing ERP, WMS, and TMS | Consolidated validation worksheet | Route validator |
| Route validator | Consolidated validation worksheet | Validate information and apply new features in the ERP | Consolidated validation | Accounting |

**Table 11.** *Cont.*

| Objective | | | |
|---|---|---|---|
| Sales force | Request for customer product returns | Manage customer product returns (reverse logistics) | Product returns | DC and Transportation |
| **Documentation** | | Consolidated orders, Picking Orders, Vehicle Loading List, Verification Form. | | |
| **Resources** | | Regional DC infrastructure and material handling equipment, vehicles, human talent, computers. | | |
| **Key Performance Indicators** | | % Service level, % freight on the sales, % of perfect deliveries, % of voided invoices | | |

The second improvement proposal consists of implementing ICTs to improve the secondary distribution. From the master plan and characterization of the secondary distribution, the importance of appropriating a TMS is established, including its synchronization with the company's WMS and ERP and other ICTs implemented in the company. As shown in Figure 2, the synchronization of multiple ICTs is required through information and technology architecture (physical and logical data, software, hardware, and communications supporting ICTs). In this sense, Table 12 describes the scope and functionality of the proposed ICT architecture based on the ERP, WMS, order management system, and TMS.

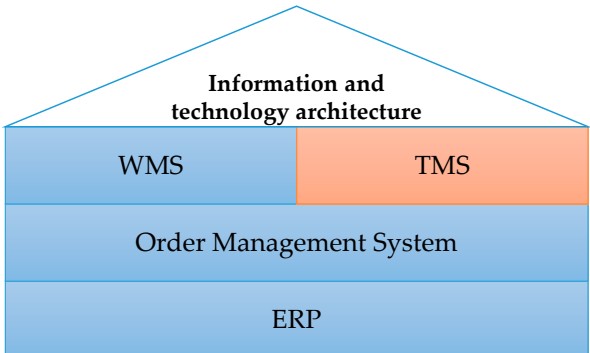

**Figure 2.** Information and technology architecture for the secondary distribution.

**Table 12.** ICT in the business architecture of the secondary distribution.

| ICT | Description |
|---|---|
| ERP | It manages company processes, including interactions with other logistics processes in the supply chain. The main functionalities are financial management, accounting, production, human talent, maintenance, and their interactions with logistics and sales. |
| OMS | ICT specialized in sales management, covering functionalities such as order taking, traceability, and control of customer compliance. It allows the design of vendor routes, control of visits, and configuration of geo-fences. |
| WMS | It allows the management of regional DCs covering operations such as reception, put-away, storage, order picking, and dispatch. It includes the management inventory transfer between the factory DC and regional DCs and between regional DCs; real-time inventory; integration of order management with sales and secondary distribution. |
| TMS | It supports the planning, execution, and control of the secondary distribution considering functions such as cargo planning, vehicle selection according to the characteristics of the orders and logistical conditions, design of routes with minimal tardiness considering city logistics, visibility and traceability of orders in the secondary distribution (from dispatch to delivery at the customer's location), management of reports and indicators, sales synchronization, billing, cost, and freight management. |

Since the TMS is the most relevant ICT in the business architecture for the logistics processes in the company under study, Table 13 describes each of its functionalities, impacts, and software and hardware required to support it.

**Table 13.** TMS functionalities.

| TMS Module | Functional Characteristics | Expected Impacts | Software/Hardware |
|---|---|---|---|
| Cargo transportation planning | Transport volume forecast: Determines the necessary transport capacity for any need for transport services. Establishment of a standardized carrier program. Scheduling of pick-up and delivery appointments for the medium term. Decision making is based on the capacities of the carriers and the demands of the clients. Generation of transport documents. | Reduce between 50 and 70% of unexpected deliveries. Increase in continuous cargo movements with a dedicated fleet between 4 to 8%. Increase in the vehicle occupancy rate between 90 and 100% of the maximum allowed capacity. | Interfaces with the information systems of transport providers and sales force. WMS software Servers. Scanning devices (RFID and/or barcode). Weight and volume measuring devices. |
| Selection of vehicles and carriers | Tender management: Preparation and creation of tenders for the contracting of transport for the medium and long term. Negotiation, management of incentives. Creation and monitoring of cargo transport contracts. Selection of transport service providers and vehicle type according to the orders to be delivered and the agreed logistical conditions. | Increase in agility for the transportation selection process from 5 to 15%. Reduction of freight transportation costs by negotiation and appropriate selection between 3% and 8%. | Interfaces with transport provider information systems. WMS software. Servers. |
| Routing | Consolidation of multiple shipping and delivery sites to optimize cargo volumes, personnel, and vehicle use. Planning the dispatch of goods. Geographic mapping and creation of efficient routes. Optimization of delivery routes through dynamic algorithms. Route planning for the available vehicles to minimize delivery times and distances traveled. Management of the vehicle fleet. | Increased efficiency in operations from 2 to 5%. Reduction of required freight between 5% to 10%. Reduction of carrier downtime between 20% to 40%. Reduction of average times of routes between 10 to 15%. Reduction of delivery time variability between 50% to 65%. | Scanning devices (RFID and/or barcode). Weight and volume measuring devices. Maps update and georeferencing. |
| Cargo visibility and traceability | Mobile asset management. Real-time visibility of merchandise in transit, vehicles, facilities, and drivers. Geographical traceability. Web collaboration with carriers. Management of load units. Traceability and monitoring of events and manipulations of the units. | 100% in-transit cargo tracking. Improvement between 10 and 15% in distribution service levels. | Scanning devices (RFID and/or barcode). GPS/GPRS devices in vehicles. |
| Visibility and traceability of orders in the secondary distribution | Real-time visibility and traceability of orders in the secondary distribution from dispatch to delivery at the customer's premises. Customer order management: Order taking, registration, tracking, and delivery. Management of deliveries and collections. Claims management. Generation of merchandise delivery notification notices to customers (ASN—Advice Shipping Notice) with stipulated dates. | Decreased order processing time from 8 to 12%. Reduction of customer returns for logistical reasons up to a value of 1%. Reduction of Voided invoices between 50 to 70% for logistical transport reasons. | Scanning devices (RFID and/or barcode). GPS/GPRS devices in vehicles. Communication terminals in vehicles. |
| Synchronization of sales, billing and logistics | Synchronization of operations between warehouses (DCs) and transportation. Financial and accounting interfaces with accounts receivable, accounts payable, and accounting ledger. Automatic freight settlement. Management of contracts and invoices with customers and suppliers. | Reduction of invoice processing time between 50 and 70%. | ERP system. Integration kit and adapters for ERP. |

**Table 13.** *Cont.*

| TMS Module | Functional Characteristics | Expected Impacts | Software/Hardware |
|---|---|---|---|
| Cost and freight management | Management of transport costs and freight calculations. Asset balance. Assigning costs to resources, activities, and cost objects. Cost estimation until the landing of merchandise. Analysis of margins and profitability. Cargo cost settlement for accounting, payments, billing. | Reduction in secondary transportation costs between 14 to 34%. | Interfaces with financial and accounting systems. Interfaces with the supplier systems, transport, and sales force. |
| Performance indicators cube | Construction of operational KPIs, statistics, Ad Hoc reports, and centralized dashboards. Real-time performance measurement, based on KPIs. Evaluation of transport service providers. Tracking the use of vehicles and assets. | Identification of critical points Transportation cost management Reduction of reaction time for decision making from 40% to 60%. | Interfaces with financial and accounting systems. Integration kit and adapters for ERP. Office automation packages. |

Key factors are pivotal for the proper implementation of the TMS. In this sense, the commitment of senior management, work teams to implement the project, effective communication among project stakeholders, and proper training of team members according to their role facilitate overcoming obstacles when they appear. Therefore, the proper implementation and integration of a TMS will support the secondary distribution operation to ensure satisfactory quality and productivity levels, standardized, and controlled processes to prevent operational and financial errors. The TMS will offer optimization tools adapted to customers, suppliers, distribution channels, and operations. All this will generate appropriate and controlled costs leading to a customer service improvement and a reduction in total transportation and distribution costs.

*4.2. Economic and Productivity Evaluation for Secondary Distribution Improvements (Stage 5)*

Table 14 quantifies the investments for the successful implementation of a TMS in the company. The improvement opportunities presented above support these investments. Hence, the acquisition and proper use of TMS require one-time investments (incurred once) and periodic investments (repeated every certain period) to acquire the technology for the secondary distribution and guarantee its operation over time.

**Table 14.** Investments for the implementation of a TMS.

| Investment | Value USD | Frequency |
|---|---|---|
| Purchasing TMS Software | 26,667 | Once |
| TMS updates (modules, applications, features) | 6667 * | Annual |
| Integration of TMS with ERP and WMS | 13,333 | Once |
| Design of the master plan and redesign of the characterization of the secondary distribution | 8533 | Once |
| Customization of TMS performance measures and reports | 7467 | Once |
| Integration with transport service providers (GPS Tracker, mobile routing device) | 66,667 | Once |
| Acquisition of servers and computer support equipment for the TMS | 5333 | Once |
| Lead TMS Implementation and Tracking Analyst | 11,200 | Annual |
| TMS technical services and contingencies | 5333 | Annual |
| Services to transmit ASNs (Advance Shipping Notice) | 24,000 | Annual |
| Training and staff updates in the management of TMS | 1600 | Annual |

* The software purchase does not include updates. These must be done as new modules and applications appear in the TMS.

According to Table 13, which shows the expected impacts from the implementation of the TMS and based on the information regarding the quantitative measurement of the performance of the secondary distribution, Table 15 presents the expected productivity impacts supported by Table 13.

**Table 15.** Productivity impacts generated by the implementation of the TMS.

| Factor | Current Value | Expected Impacts | Expected Value USD | Saving USD |
|---|---|---|---|---|
| Voided invoices | Voided invoices worth USD 115,399 due to receipt time failure, driver delivery error, and driver not visiting customer. | Voided invoices reduction of 60% for transport logistics reasons. | 69,239 | 46,159 |
| Vehicle utilization | 61% vehicle occupancy representing a secondary freight cost of USD 1,592,236. | Vehicle occupancy rate increases to 90% of the maximum allowed capacity (75%). | Vehicle occupancy of 67.5% increases the use of vehicles by 6.5%, saving required freight. | 103,495 |
| Freight contracting costs | USD 1,488,741 (Total cost subtracting savings for vehicle utilization) | A 5% increase in continuous cargo movements with a dedicated fleet, a 60% reduction in invoice processing time, and a 20% reduction in carrier downtime allow negotiating a 3% reduction in freight contract costs with carriers. | 1,444,079 | 44,662 |
| Routing Time | USD 1,444,079 USD (Total cost subtracting savings for vehicle utilization and savings from freight contracts) | 100% in-transit cargo tracking, scheduling, and dynamic routing reduce average route times by 10%. | 1,299,671 | 144,408 |
| Customer service | 85.5% customer service level | 4% increase in operations efficiency, 50% reduction in delivery time variability, 10% decrease in order processing time. In addition to this, the impacts generated in voided invoices due to distribution logistics cause a 9.5% improvement in distribution service levels. | 95% customer service level generating a buyback or increase in sales in current customers equivalent to 3%, to which a 4% profit factor is applied. | 31,079 |

From Table 15, the increase in customer service generates an increase in the frequency and volumes of customer orders. The customer service can increase the purchase volume of current customers by at least 3%. For practical purposes, the estimations consider a 4% profit margin for the sold products to quantify the profit generated by the TMS in terms of increased purchases or repurchases. Therefore, the company's annual sales represent USD 25,899,320, of which an increase of 3% equals USD 776,980, providing a supplemental income equivalent to USD 31,079 (4% profit margin) due to good customer service. Moreover, the TMS implementation improves the transportation selection process, reduces the number of unexpected and urgent deliveries, and reduces the response time for decision making, reflected in increased efficiency in the transportation and distribution processes.

As shown in Table 16, the secondary distribution costs represent 4.72% on annual sales, with which the goals established and proposed by the company are met. In addition, the implementation of TMS allows for reducing the cost of voided invoices caused by secondary distribution from 0.45% to 0.27% on annual sales. Likewise, the improvements in voided invoices, efficiency in the distribution operation, reduction of the variability of the delivery time, and reduction of the order processing time allow for reaching the desired service level, from 85.5% to 95%.

**Table 16.** Impact values generated by the implementation of the TMS.

| Impacts | Value |
|---|---|
| Total savings in voided invoices | USD 46,159 |
| Total savings in secondary distribution | USD 292,565 |
| Total income for good customer service | USD 31,079 |
| Expected value of secondary distribution | USD 1,222,433 |

Table 17 presents the economic analysis to determine the feasibility of the TMS project. The economic evaluation is based on years to visualize the implementation time of the TMS (stipulated in 24 months) and the investment return over time. Likewise, the economic analysis considers annual inflation of 5% to update the investments values and productivity savings. As shown in Table 17, the TMS implementation takes two years; then, the recovery of the investment begins after this period due to the impacts of the project. This information provides an objective analysis of the project's economic impact to support the decision to implement a TMS to improve the efficiency of the secondary distribution.

**Table 17.** Economic analysis of the TMS implementation.

| Concept. | Year (Amount in USD) | | | | | |
|---|---|---|---|---|---|---|
| | 0 | 1 | 2 | 3 | 4 | 5 |
| **Incomes** | | | | | | |
| Voided invoices | 0 | 0 | 46,159 | 48,467 | 50,891 | 53,435 |
| Vehicle utilization | 0 | 0 | 103,495 | 108,670 | 114,104 | 119,809 |
| Freight contracting costs | 0 | 0 | 44,662 | 46,895 | 49,240 | 51,702 |
| Routing time | 0 | 0 | 144,408 | 151,628 | 159,210 | 167,170 |
| Customer service | 0 | 0 | 31,079 | 32,633 | 34,265 | 35,978 |
| Total Income | 0 | 0 | 369,803 | 388,293 | 407,710 | 428,094 |
| **Expenses** | | | | | | |
| Purchase TMS Software | 26,667 | 0 | 0 | 0 | 0 | 0 |
| TMS updates (modules, applications, features) | 0 | 0 | 6667 | 7000 | 7350 | 7718 |
| Integration of TMS with ERP and WMS | 13,333 | 0 | 0 | 0 | 0 | 0 |
| Design of the master plan and redesign of the characterization of the secondary distribution | 8533 | 0 | 0 | 0 | 0 | 0 |
| Customization of TMS performance measures and reports | 7467 | 0 | 0 | 0 | 0 | 0 |
| Integration with transport service providers (GPS Tracker, mobile routing device) | 66,667 | 0 | 0 | 0 | 0 | 0 |
| Acquisition of servers and computer support equipment for the TMS | 5333 | 0 | 0 | 0 | 0 | 0 |
| TMS implementation and tracking analyst | 11,200 | 11,760 | 2348 | 12,965 | 3614 | 14,294 |
| TMS technical services and contingencies | 5333 | 5600 | 5880 | 6174 | 6483 | 6807 |
| Services to transmit ASNs (Advance Shipping Notice) | 24,000 | 25,200 | 6460 | 27,783 | 29,172 | 30,631 |
| Training and staff updates in the management of TMS | 1600 | 1680 | 1764 | 1852 | 1945 | 2042 |
| Total Expenses | 170,133 | 44,240 | 23,119 | 55,774 | 48,564 | 61,492 |
| Net Profit | −170,133 | −44,240 | 346,684 | 332,519 | 359,146 | 366,602 |

Table 18 shows the economic impact indicators such as the net present value (NPV), the internal rate of return (IRR), the investment payback period (IPP), and the return on investment (ROI) calculated from Table 16. The NPV uses a 20% annual rate, representing an attractive rate for the company. The NPV is USD 659,295, which is equivalent to an IRR of 84%; that is, the TMS implementation project covers its investment and operational costs, generating additional value in 5 years, and with a return rate higher than four times the investor opportunity rate (discount rate).

**Table 18.** Economic variables in the TMS implementation.

| Economic variable | Value |
|---|---|
| Discount Rate | 20% |
| NPV | USD 659,295 |
| IRR | 84% |
| IPP | 2 years, 9 months |

Likewise, the project shows that the IPP is two years and nine months, establishing that once the TMS is fully operational, it covers its investment costs in just nine months of operation. Regarding the return on investment (ROI), this determines the economic benefit generated by the TMS system. The ROI results from the division between the net income of the TMS implementation and implementing costs. Table 19 shows that the project does not provide ROI for the first two periods when the TMS is being implemented. Then, the TMS implementation provides ROI of 38%, 135%, 205%, and 260% for years 2, 3, 4, 5, respectively.

**Table 19.** ROI for different periods.

| Concept | Year (Amount in USD) | | | | | |
|---|---|---|---|---|---|---|
| | 0 | 1 | 2 | 3 | 4 | 5 |
| Accumulated investments (USD) | 170,133 | 214,373 | 267,492 | 323,267 | 381,830 | 443,321 |
| Profits (USD) | −170,133 | −214,373 | 102,312 | 434,832 | 783,978 | 1,150,581 |
| ROI | −100% | −100% | 38% | 135% | 205% | 260% |

This information confirms the TMS is a viable solution to guarantee the sustainability and growth of the secondary distribution logistics operations of the company. Therefore, the benefits of TMS come from the improvements offered by the modules of cargo transport planning, vehicle and carrier selection, routing, visibility and traceability for cargo and orders, sales synchronization, billing and logistics, cost and freight management of the secondary distribution and management indicators cube. Consequently, the proposed DMAIC methodology technically and economically validated a TMS for a distribution system, demonstrating the achievement of company objectives related to the reduction of delivery time variability, order processing time, voided invoices, distribution costs, the increase in customer service, and efficiency in the distribution operation, as well as generating profitability for the medium and long term.

### 4.3. Discussion

From the traditional DMAIC methodology, adaptable methodologies can arise to solve multiple logistics problems involving ICT. In this sense, by proposing the assess element and replacing the control element, the DMAIS (define–measure–analyze–improve–assess) methodology is obtained, which, unlike other technology assessment methods, involves the analysis of the entire logistics process. This approach allows for analyzing the requirements of the process, considering the relevance of implementing, modifying, or

improving technologies, and selecting the most appropriate technologies that provide the solutions required and expected in the process.

Consequently, the define phase determined the problem of the secondary distribution, the current value, and the desired value of related indicators. The measure phase was based on CTL factors represented in five variables that allowed us to understand the current problem. The analysis phase provided a descriptive diagnosis and a quantitative performance measurement for the secondary distribution to understand the characteristics and performance of the process. The improve phase identified and selected improvement opportunities to meet the goals set for the secondary distribution, prioritizing the secondary distribution management, and proposing a master plan and characterization for the secondary distribution, highlighting the importance of appropriating a TMS is established, including its synchronization with the company's WMS and ERP, and other ICTs implemented in the company. The assess phase validated the investments, productivity impacts, savings, incomes, and economic analysis generated by the implementation of the TMS.

In the case of the company under study, the secondary distribution process presented relevant challenges in terms of CTL factors (secondary freight expenses, voided invoices, product quality returns, product turnover returns, and service level). A survey conducted to employees involved in the strategic, tactical, and operational levels allowed for obtaining insights about the CTL factors of the secondary distribution process. The structured dialogue methodology prioritized the improvement opportunities, highlighting a master plan and a SIPOC characterization, which suggested the adoption of a TMS in sync with the WMS and the ERP. The expected benefits for the company related to the adoption of a TMS directly impact the CTL, offering savings in voided invoices and secondary distribution, and incomes for good customer service, providing profits and ROI after two years of TMS operation. Moreover, the commitment of the top management is pivotal to ensure the success of the DMAIS methodology. Top management must support improvement projects by providing resources and personnel for the diagnostic and definition of the problem, analysis of information and indicators, creation of improvement opportunities, design of a master plan and SIPOC characterization of the logistics process to be improved, and economic and productive evaluation of the selected project.

As has been done in other investigations [36,55,59,62,63], this study focuses on a case study to demonstrate the efficiency and effectiveness of the DMAIS methodology by establishing the main problems related to secondary distribution, the causes, and solutions to these problems. In this case, a TMS is significant due to the potential costs and benefits generated to the company. Specifically, the implementation of a TMS generates expected impacts in reducing unexpected deliveries, freight transportation costs, increased efficiency in operations, in-transit cargo tracking, improvement in service level, decreased order processing time, reduction of invoice processing time, reduction in secondary transportation costs, among others. These impacts allow for reaching the operational and strategic goals of the organization and supply chain.

## 5. Conclusions

This study proposed a plan to improve the secondary distribution for a company in the productive sector, thereby allowing increases to customer services, reducing the value of freight over sales, and reducing customer returns to guarantee the operational and economic sustainability of the company. DMAIC methodology was implemented by replacing the control component with the assess component to perform the economic and productivity evaluation of the implementation of a TMS since the improvement proposals were formulated and evaluated. This methodology defined the main problems of the secondary distribution, facilitating the understanding of the performance and characteristics of the process based on primary and secondary information. From the proposed improvement opportunities, the most relevant and appropriate solutions were selected using a structured dialogue. This methodology prioritized the design of a master plan for the secondary distribution and the characterization of the secondary distribution process.

This study defined the principal ICTs that should compose the business architecture of the secondary distribution, with emphasis on the TMS as a technological tool of significant impact and relevance for planning, execution, and control of the distribution process. The economic analysis of the TMS implementation showed that this technology allows for achieving the objectives set by the company, providing a secondary distribution cost of 4.72% on annual sales. Additionally, TMS reduces the cost of voided invoices caused by the secondary distribution from 0.45% to 0.27%. Likewise, the desired service level could increase from 85.5% to 95% through improvements in voided invoices, efficiency in the distribution operation, reduction in delivery time variability, and reduction in order processing time.

Moreover, the economic analysis showed that the TMS implementation is highly profitable for the medium and long term. Therefore, the proposed methodology validates the technical, productive, and economic viability for ICT implementations in logistics processes, offering a solution to improve logistics problems such as the secondary distribution process. The limitations of this research are mainly linked to the use of a case study in a single company, and the proposed methodology does not cover the post-implementation phase to compare the expected performance to the performance obtained after the improvement of the secondary distribution process. Another limitation of the study is associated with the supplier selection process for the ICT; then, future research may integrate a supplier selection method (a multicriteria decision-making method) to the proposed technology and compare the expected results versus the actual results to measure the precision of the methodology. Likewise, future works should implement the proposed methodology to validate other ICT implementations in logistics processes, such as purchasing, demand management, scheduling, and warehouse management.

**Author Contributions:** Conceptualization, J.A.C. and R.A.G.; methodology, R.A.G.; validation, J.A.C. and R.A.G.; formal analysis, R.A.G.; investigation, J.A.C. and R.A.G.; resources, R.A.G. and P.C.; writing—original draft preparation, J.A.C. and R.A.G.; writing—review and editing, J.A.C. and P.C.; visualization, J.A.C. and R.A.G.; supervision, J.A.C. and P.C. All authors have read and agreed to the published version of the manuscript.

**Funding:** This research received no external funding.

**Institutional Review Board Statement:** Not applicable.

**Informed Consent Statement:** Not applicable.

**Data Availability Statement:** The data presented in this study are available on request from the corresponding author.

**Conflicts of Interest:** The authors declare no conflict of interest.

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
