# Peer review of "ICT Validation in Logistics Processes: Improvement of Distribution Processes in a Goods Sector Company"

_informatics, doi:10.3390/informatics8040075_

Round 1
Reviewer 1 Report
The introduction does not provide sufficient background. The introduction does not include all relevant references. The authors used inappropriate self-citation (reference no. 27). The research gap is not identified and supported by the literature. The whole manuscript is very weak from the perspective of scientific soundness and significance of content. The manuscript does not contain any significant scientific progress, as these are only one case study processed in one company. This tends to lead to an interpretive case study. The manuscript does not contain the limits of research. The manuscript does not contain a discussion.
Author Response
Dear Abigail Zhao Medellín, October 25th, 2021
Informatics
MDPI
We communicate with you through this letter in order to outline every change made in the article informatics-1370975 “ICT Validation in Logistics Processes: Improvement of Distribution Processes in a Goods Sector Company” based on the observations and suggestions provided by the reviewers (Reviewer 1, Reviewer 2, Reviewer 3, Reviewer 4). All changes and modifications made to the manuscript are highlighted in green (new paragraphs) and blue (paragraph relocated) in the revised version.
Comments from author to reviewers:
-Reviewer 1
The introduction does not provide sufficient background. The introduction does not include all relevant references.
We thank the reviewers for their comments and suggestions, which allowed us to substantially improve the quality and contribution of the manuscript.
R// A new section called Research Background is created to provide sufficient background to the manuscript (Lines 126-231)
R// New references are added in the Introduction and Research Background:
Acero, R., Torralba, M., Pérez-Moya, R., & Pozo, J. A. (2020). Value stream analysis in military logistics: The improvement in order processing procedure. Applied Sciences (Switzerland), 10(1), 106. https://doi.org/10.3390/app10010106
Basios, A., & Loucopoulos, P. (2017). Six Sigma DMAIC Enhanced with Capability Modelling. In D. D. T. B. Manolopoulos Y. Zaki M. (Ed.), Proceedings - 2017 IEEE 19th Conference on Business Informatics, CBI 2017 (Vol. 2, pp. 55–62). Institute of Electrical and Electronics Engineers Inc. https://doi.org/10.1109/CBI.2017.70
Bates, O., Friday, A., Allen, J., McLeod, F., Cherrett, T., Wise, S., Piecyk, M., Piotrowska, M., Bektas, T., & Nguyen, T. (2018). ICT for Sustainable Last-Mile Logistics: Data, People and Parcels. ICT4S2018: 5th International Conference on Information and Communication Technology for Sustainability, 52, 49–67. https://doi.org/10.29007/67nl
Cano, J. A., Salazar, F., & Gómez-montoya, R. A. (2020). ICT Validation Methodologies for Logistics Management. International Journal of Supply Chain Management, 9(5), 1157–1163. https://ojs.excelingtech.co.uk/index.php/IJSCM/article/view/5571
Cano, J. A., Salazar, F., Gómez-montoya, R. A., & Cortés, P. (2021). Disruptive and Conventional Technologies for the Support of Logistics Processes: A Literature Review. International Journal of Technology, 12(3), 448–460. https://doi.org/10.14716/ijtech.v12i3.4280
De Marco, A., Mangano, G., Zenezini, G., Cagliano, A. C., Perboli, G., Rosano, M., & Musso, S. (2017). Business Modeling of a City Logistics ICT Platform. Proceedings - International Computer Software and Applications Conference, 2, 783–789. https://doi.org/10.1109/COMPSAC.2017.76
Dossou, P.-E., Foreste, L., & Misumi, E. (2021). Intelligent Support System for Healthcare Logistics 4.0 Optimization in the Covid Pandemic Context. Journal of Software Engineering and Applications, 14(06), 233–256. https://doi.org/10.4236/jsea.2021.146014
Evangelista, P., Santoro, L., & Thomas, A. (2018). Environmental sustainability in third-party logistics service providers: A systematic literature review from 2000-2016. Sustainability (Switzerland), 10(5), 1627. https://doi.org/10.3390/su10051627
Gutierrez-Gutierrez, L., de Leeuw, S., & Dubbers, R. (2016). Logistics services and Lean Six Sigma implementation: a case study. International Journal of Lean Six Sigma, 7(3), 324–342. https://doi.org/10.1108/IJLSS-05-2015-0019
Hankel, A. (2012). Transport optimization project team uses DMAIC to improve efficiency, customer satisfaction. Quality Progress, 45(11), 30–36. https://www.scopus.com/inward/record.uri?eid=2-s2.0-84870658790&partnerID=40&md5=11641636316d0555e2cf2c477e680929
Kiisler, A., Solakivi, T., & Hilmola, O. P. (2020). Supply chain and ICT issues of Estonia: Survey findings. Procedia Computer Science, 176, 828–837. https://doi.org/10.1016/j.procs.2020.09.078
Kimanthi, I. (2016). Framework for government ICT disruptive innovation projects: a case study of digital TV migration in Kenya [Strathmore University]. http://su-plus.strathmore.edu/handle/11071/4774%0AThis
Ko, A., Vas, R., Kovacs, T., & Szabo, I. (2019). Knowledge creation from the perpective of the supply chain. The role of ICT. Society and Economy, 41(3), 311–329. https://doi.org/10.1556/204.2019.009
Korczak, J., & Kijewska, K. (2019). Smart Logistics in the development of Smart Cities. Transportation Research Procedia, 39(2018), 201–211. https://doi.org/10.1016/j.trpro.2019.06.022
Kumar, S., Dieveney, E., & Dieveney, A. (2009). Reverse logistic process control measures for the pharmaceutical industry supply chain. International Journal of Productivity and Performance Management, 58(2), 188–204. https://doi.org/10.1108/17410400910928761
Mensah, P., Merkuryev, Y., & Longo, F. (2015). Using ICT in developing a resilient supply chain strategy. In S. M. Ginters E. (Ed.), Procedia Computer Science (Vol. 43, Issue C, pp. 101–108). Elsevier. https://doi.org/10.1016/j.procs.2014.12.014
Nicoletti, B., Nicoletti, B., & Vergata, T. (2013). Lean Six Sigma and digitize procurement. International Journal of Lean Six Sigma, 4(2), 184–203. https://doi.org/10.1108/20401461311319356
Omotayo, A., & Melan, M. (2017). Factors influencing the Information and Communication Technology (ICT) of third party logistics in Malaysia. International Journal of Supply Chain Management, 6(2), 202–208.
Panayiotou, N. A., & Stergiou, K. E. (2021). A systematic literature review of lean six sigma adoption in European organizations. International Journal of Lean Six Sigma, 12(2), 264–292. https://doi.org/10.1108/IJLSS-07-2019-0084
Ul-Hameed, W., Shabbir, M. S., Imran, M., Raza, A., & Salman, R. (2019). Remedies of low performance among pakistani E-logistic companies: The role of firm’s IT capability and information communication technology (ICT). Uncertain Supply Chain Management, 7(2), 369–380. https://doi.org/10.5267/j.uscm.2018.6.002
Wang, Y., Rodrigues, V. S., & Evans, L. (2015). The use of ICT in road freight transport for CO 2 reduction – an exploratory study of UK’s grocery retail industry. The International Journal of Logistics Management, 26(1), 2–29. https://doi.org/10.1108/IJLM-02-2013-0021
Waseem-Ul-Hameed, Nadeem, S., Azeem, M., Aljumah, A. I., & Adeyemi, R. A. (2018). Determinants of e-logistic customer satisfaction: A mediating role of information and communication technology (ICT). International Journal of Supply Chain Management, 7(1), 105–111.
The authors used inappropriate self-citation (reference no. 27).
R// The citation mentioned by reviewer # 1 was removed.
The research gap is not identified and supported by the literature.
R// The following paragraphs are added to support the research gap that is filled with this study (Lines 77-117):
Considering that the success of the ICT implementation in logistics processes requires proper selection and validation [30], that the selection of an appropriate ICT has a significant impact on the competitive advantage of logistics companies and logistics operations [18], and that companies make investments acquiring, updating, and maintaining technological infrastructure [31], it is necessary to review the appropriate methodologies to solve structured problems and support the selection and implementation of ICT for the secondary distribution.
Methodologies like Technology Acceptance Model (TAM), Unified Theory of Acceptance and Use of Technology (UTAUT), Dialogue-Access-Risk Assessment-Transparency (DART), Define-Measure-Analyze-Improve-Control (DMAIC), Failure Modes and Effects Analysis (FMEA), and Quality Function Deployment (QDF) can predict the success of an ICT according to user perceptions, associated risks, and process modifications, ensuring ICTs become facilitators of value creation in logistics management. From these methodologies, the DMAIC is the most characteristic methodology of lean six sigma implementation in organization processes [32] and is usually used to support and verify the ICT appropriation, solving structured problems, supporting management processes in ICT implementation projects [33,34], and guaranteeing a successful ICT implementation in logistics processes [30]. Likewise, the DMAIC methodology can be modified and supplemented, as in the case of DMADV, which employs the define, measure, analyze, design and verify phases to replace existing systems with new processes [32].
Based on the research gaps in the literature, it is identified that it is necessary to analyze in more detail the connections between lean six sigma (LSS) and ICT [35], extend the understanding of LSS implementation in specific contexts to create a theoretical and empirical background that better help implementing LSS in logistics services environments successfully [36], produce more papers using other already existing or completely new methods related to DMAIC [32], and research real organizations in collaboration with ICT platforms [37]. Therefore, this study aims to improve the secondary distribution process in mass consumer companies through the use and appropriation of ICT, offering an improvement approach based on data analysis and management tools in supply chain logistics, especially for the distribution process improvement in terms of service level, freight costs on sales, and returns from customers. The main contributions of this paper are as follows:
- The DMAIC methodology is used to improve a secondary distribution process that involves the selection and evaluation of ICT and promotes the connections between LSS and ICT in logistics.
- The DMAIC methodology is modified by discarding the control component since the improvement proposals are formulated, and it is replaced by the assess component to evaluate the economic and productive impacts generated in the secondary distribution system of a company.
- This study provides an application in a large food company to show the efficiency and effectiveness of the proposed methodology.
The whole manuscript is very weak from the perspective of scientific soundness and significance of content.
R // The manuscript was reinforced from the perspective of scientific soundness through the addition of multiple paragraphs in the Introduction, Research Background, and Discussion. Please see the following lines:
- Lines 55–67
- Lines 77–117
- Lines 146–149
- Lines 183–196
- Lines 202–204
- Lines 217–231
- Lines 562–595
- Lines 622–629
The manuscript does not contain any significant scientific progress, as these are only one case study processed in one company. This tends to lead to an interpretive case study.
R// A case study was carried out in a food company to test the benefits of the proposed methodology, as has been done in previous studies [37,56,60,63,64]
- Gutierrez-Gutierrez, L.; de Leeuw, S.; Dubbers, R. Logistics services and Lean Six Sigma implementation: a case study. Int. J. Lean Six Sigma 2016, 7, 324–342, doi:10.1108/IJLSS-05-2015-0019.
- Senthilkumar, T.; Karthi, S.; Devadasan, S.R.; Sivaram, N.M.; Sreenivasa, C.G.; Murugesh, R. Implementation of DMAIC methodology in supply chains to reduce customer end-rejections: A case study in an Indian SME. Int. J. Product. Qual. Manag. 2012, 10, 388–409, doi:10.1504/IJPQM.2012.048755.
- Mishra, P.; Sharma, R.K. Measuring business performance in a SCN using Six Sigma methodology-a case study. Int. J. Ind. Syst. Eng. 2017, 25, 76–109, doi:10.1504/IJISE.2017.080689.
- Hankel, A. Transport optimization project team uses DMAIC to improve efficiency, customer satisfaction. Qual. Prog. 2012, 45, 30–36.
- Acero, R.; Torralba, M.; Pérez-Moya, R.; Pozo, J.A. Value stream analysis in military logistics: The improvement in order processing procedure. Appl. Sci. 2020, 10, 106, doi:10.3390/app10010106.
R// The contributions made by the study are specified in the Introduction in a new paragraph (Lines 97-117):
Based on the research gaps in the literature, it is identified that it is necessary to analyze in more detail the connections between lean six sigma (LSS) and ICT [35], extend the understanding of LSS implementation in specific contexts to create a theoretical and empirical background that better help implementing LSS in logistics services environments successfully [36], produce more papers using other already existing or completely new methods related to DMAIC [32], and research real organizations in collaboration with ICT platforms [37]. Therefore, this study aims to improve the secondary distribution process in mass consumer companies through the use and appropriation of ICT, offering an improvement approach based on data analysis and management tools in supply chain logistics, especially for the distribution process improvement in terms of service level, freight costs on sales, and returns from customers. The main contributions of this paper are as follows:
- The DMAIC methodology is used to improve a secondary distribution process that involves the selection and evaluation of ICT and promotes the connections between LSS and ICT in logistics.
- The DMAIC methodology is modified by discarding the control component since the improvement proposals are formulated, and it is replaced by the assess component to evaluate the economic and productive impacts generated in the secondary distribution system of a company.
- This study provides an application in a large food company to show the efficiency and effectiveness of the proposed methodology.
The manuscript does not contain the limits of research.
R // A paragraph was added to establish the limitations of the study (Lines 622-629):
The limitations of this research are mainly linked to the use of a case study in a single company, and the proposed methodology does not cover the post-implementation phase to compare the expected performance to the performance obtained after the improvement of the secondary distribution process. Another limitation of the study is associated with the supplier selection process for the ICT, for which future research may integrate a sup-plier selection method (a multicriteria decision-making method) to the proposed technology and compare the expected results versus the actual results to measure the precision of the methodology.
The manuscript does not contain a discussion.
R // Section “4.3 Discussion” was created, showing there the contributions of the proposed methodology and its importance to prioritize the implementation of technologies, especially in companies that have limited resources and multiple logistical needs to cover (Lines 562- 595):
From the traditional DMAIC methodology, adaptable methodologies can be obtained to solve multiple business problems, such as improvement problems in logistics involving ICT. In this sense, by proposing the Assess element and replacing the Control element, the DMAIS (Define-Measure-Analyze-Improve-Assess) methodology is obtained, which, unlike other technology assessment methods, involves the analysis of the entire logistics process. This approach allows analyzing the requirements of the process, considering the relevance of implementing, modifying, or improving technologies, and selecting the most appropriate technologies that provide the solutions required and expected in the process.
Consequently, the Define phase determined the problem of the secondary distribution, the current value, and the desired value of related indicators. The Measure phase was based on CTL factors represented in five variables that allowed us to understand the current problem. The Analysis phase provided a descriptive diagnosis and a quantitative performance measurement for the secondary distribution to understand the characteristics and performance of the process. The Improve phase identified and selected improvement opportunities to meet the goals set for the secondary distribution, prioritizing the secondary distribution management, and proposing a master plan and characterization for the secondary distribution, highlighting the importance of appropriating a TMS is established, including its synchronization with the company's WMS and ERP, and other ICTs implemented in the company. The Assess phase validated the investments, productivity impacts, savings, incomes, and economic analysis generated by the implementation of the TMS.
As has been done in other investigations [36,55,59,62,63], this study focuses on a case study to demonstrate the efficiency and effectiveness of the DMAIS methodology by establishing the main problems related to secondary distribution, the causes, and solutions to these problems. In this case, a TMS is significant due to the potential costs and benefits generated to the company. Specifically, the implementation of a TMS generates expected impacts in reducing unexpected deliveries, freight transportation costs, increased efficiency in operations, in-transit cargo tracking, improvement in service level, decreased order processing time, reduction of invoice processing time, reduction in secondary transportation costs, among others. These impacts allow reaching the operational and strategic goals of the organization and supply chain.
Comments from author to reviewers:
-Reviewer 2
I consider this manuscript to be a good study of ICT in logistics. The following recommendations are my suggestions for an improvement of the paper:
The introduction seems too big. The author should reduce it and focus on the theme of the study. I do not see the rationale for separating section 1.1 from the main introduction, there is only one section. Maybe, it would be better to write a concise, short, and very focused introduction (relevance and aim), and then establish a separate Literature Review section to better and more comprehensively address the theoretical background of the topic.
We thank the reviewers for their comments and suggestions, which allowed us to substantially improve the quality and contribution of the manuscript.
R// The Introduction was reduced focusing on the subject of study, being concise and establishing the main contributions. We removed Section 1.1 from the Introduction. The “Research Background” section was created to dedicate it to the literature review and to address the background on secondary distribution, methodologies for ICT validation and on the TMS.
In the Introduction the paragraphs corresponding to the following lines were added:
- Lines 55–67
- Lines 77-117
Some of the text that was previously in the Introduction became part of the Research Background and is underlined in blue in the revised manuscript. In Research Background, the paragraphs corresponding to the following lines were added:
- Lines 146–149
- Lines 183–196
- Lines 202–204
- Lines 217–231
Explain the selection of the methodology in more detail. It would be good to give examples of previous studies that employed the DMAIC approach in studying ICT in logistics, reveal and discuss advantages and disadvantages, also discuss other approaches that exist and explain why the DMAIC was selected.
R// We added some paragraphs in Research Background to show studies that employed the DMAIC approach in studying ICT in logistics. We added Table 1 to reveal and discuss advantages and disadvantages of methodologies to validate ICT. We also explain why the DMAIC was selected. The paragraphs correspond to Lines 183-196 and Lines 217-231:
However, it is necessary to use a technology validation methodology to determine the relevance of an ICT for a logistics process. Based on [31], Table 1 presents a comparison between the main methodologies used for the validation of ICT for logistics processes, highlighting the advantages and disadvantages of each one. From these options, the DMAIC is chosen, which despite representing a generalist and not a very standardized methodology, it can be adapted and modified according to the requirements of the logistics process. Likewise, DMAIC is used because it is a comprehensive methodology that seeks the continuous improvement of processes, focusing not exclusively on establishing whether the ICT is relevant or not but also on determining the requirements to overcome identified problems. Accordingly, in this study, the Control component is replaced by the Assess component because the proposed scope covers up to the economic and productive validation of the solutions.
Table 1. Advantages and disadvantages of ICT validation methodologies.
|
Methodology |
Advantages |
Disadvantages |
|
DART |
It allows technology providers to better involve customers as collaborators, facilitates dialogue with consumers and risk assessment on both sides. It provides the developer with consumer expectations and experiences to improve trust. |
It was created for generic products or services and needs to guarantee an intense interaction with each consumer. Discussing options openly not necessarily gives customers a degree of control over the responsibilities to assume. |
|
DMAIC |
It is used as a continuous improvement method for understanding root causes of a problem, provides procedures for the effective integration of tools within a systematic framework, and includes powerful statistical techniques for hypothesis verification. |
The generality of the method. The identification of the causes of potential problems has no strategic orientation. It does not use simulation and optimization tools to model complexity. |
|
QDF |
It can transform the customer's needs into technical solutions to improve the performance of a process, covering all the development stages of a technology. |
Information about individual judgments can be generated in multiple formats that may be alien to the knowledge of the individual. The preferences generated can be difficult to assess consistently. |
|
TAM |
Model for predicting the use of information and communications technologies. Effective alternative to analyze the reasons that lead individuals to adopt new technologies. Simplicity, adaptability, and theoretical strength. |
Model dependent on external factors that can be diverse like cultural factors. Lack of relationship among psychological, social, and contextual variables such as material access conditions or digital user skills. |
|
UTAUT |
It helps to understand the acceptance factors during the proactive design of technologies, aimed at users less likely to adopt and use new systems. |
The limited application for some business areas. Most of the studies performed have been within the same country, which leaves out culture as a technology acceptance criterion. |
DMAIC can be used in various sectors because it is not a standardized procedure and is used as a continuous improvement method [33]. It has been used as a framework for supporting continuous improvement in logistics processes, encompassing logistics services [37], procurement processes with ICT for process digitization purposes [36], reverse logistics processes for the pharmaceutical industry [61], inbound logistics and production scheduling to manage the transformation from traditional operations towards the integration of the IoT and Cyber-Physical-Systems [62], service and delivery processes to increase the level of compliance with service agreements, reduce CO2 emissions, improve invoice processing time and delivery fill rates [63], supply chain strategies in collaboration with ICT platforms to make organizations resilient to disruptions [38], internal logistics and hospital sustainable digital transformation to elaborate an intelligent support system [55], and supply chain management to minimize lead times focusing on the improvement in processes, information systems, organizational structures, and advances in distribution and transportation technologies [64].
Comments from author to reviewers:
-Reviewer 3
This article is well written and tackles an interesting subject. It refers to a wide known methodology - A DMAIC based methodology - to solve problems and support technology implementation projects. It is a good idea for such implication. It is supported by real investigation what increases the value of this idea. Its literature is well selected and show the backround to the study. The conclusions are the summary of the obtained results.
R// We thank the reviewers for their comments, and we are pleased that they have highlighted the contribution of the manuscript.
Comments from author to reviewers:
-Reviewer 4
Interesting application of DMAIC methodology. Paper is sound and interesting.
R// We thank the reviewers for their comments, and we are pleased that they have highlighted the contribution of the manuscript.
Finally, we thank the reviewers for their suggestions and observations, which helped to improve the quality of the article, both in its form and content, ensuring that it is an article of high impact for the academic and scientific community.

Reviewer 2 Report
I consider this manuscript to be a good study of ICT in logistics. The following recommendations are my suggestions for an improvement of the paper:
The introduction seems too big. The author should reduce it and focus on the theme of the study. I do not see the rationale for separating section 1.1 from the main introduction, there is only one section. Maybe, it would be better to write a concise, short, and very focused introduction (relevance and aim), and then establish a separate Literature Review section to better and more comprehensively address the theoretical background of the topic.
Explain the selection of the methodology in more detail. It would be good to give examples of previous studies that employed the DMAIC approach in studying ICT in logistics, reveal and discuss advantages and disadvantages, also discuss other approaches that exist and explain why the DMAIC was selected.
Author Response

(The authors gave the same response as above.)

Reviewer 3 Report
This article is well written and tackles an interesting subject. It refers to a wide known methodology - A DMAIC based methodology - to solve problems and support technology implementation projects. It is a good idea for such implication. It is supported by real investigation what increases the value of this idea. Its literature is well selected and show the backround to the study. The conclusions are the summary of the obtained results.
Author Response

(The authors gave the same response as above.)

Reviewer 4 Report
Interesting application of DMAIC methodology. Paper is sound and interestuing.
Author Response

(The authors gave the same response as above.)

Round 2
Reviewer 2 Report
My recommendations have been addressed
Author Response
Dear Abigail Zhao Medellín, October 28th, 2021
Informatics
MDPI
We communicate with you through this letter in order to outline every change made in the article informatics-1370975 “ICT Validation in Logistics Processes: Improvement of Distribution Processes in a Goods Sector Company” based on the observations and suggestions provided by the reviewer (Reviewer 2). All changes and modifications made to the manuscript are highlighted in yellow (English writing and grammar corrections) in the revised version.
Comments from author to reviewers:
-Reviewer 2
My recommendations have been addressed.
We are pleased that all the reviewer's recommendations have been satisfactorily addressed.
Likewise, the entire document was revised, and the English writing and grammar were improved to facilitate the reading of the manuscript.
Finally, we thank the reviewers for their suggestions and observations, which helped to improve the quality of the article, both in its form and content, ensuring that it is an article of high impact for the academic and scientific community.
